



# Interdisciplinary Pressure Cooker: environmental risk communication skills for the next generation

Lydia Cumiskey[1,2], Matthew Lickiss[3], Robert Šakić Trogrlić[1,4] Javed Ali[1,5]

[1] Water Youth Network, Delft, 2600  MH, The Netherlands
[2] Flood Hazard Research Centre, Middlesex University, The Burroughs, Hendon, London NW4 4BT, UK
[3] Department of Typography & Graphic Communication, University of Reading, Reading RG6 6UR, UK
[4] School of Energy, Geoscience, Infrastructure and Society, Heriot-Watt University, Edinburgh EH14 4AS, UK
[5] French National Centre for Scientific Research (CNRS), Institute for Geosciences and Environmental Research (IGE), Université Grenoble Alpes, Grenoble, 38000 France

*Correspondence to*: Lydia Cumiskey (l.cumiskey@wateryouthnetwork.org)

**Abstract.** This article presents a 'Pressure Cooker' approach for building interdisciplinary risk communication capacity in young researchers and professionals through an intensive 24-hour workshop. The event successfully brought together 35 participants from around the world to work on real-life environmental hazard/risk communication challenges for two areas in Mexico. Participants worked in interdisciplinary teams, following a three-step iterative process, with support from mentors and a range of specialists to develop risk communication outputs. Feedback surveys indicate that the workshop met its goal of improving participants' knowledge of risk communication and interdisciplinary working. The workshop resulted in an interdisciplinary community of researchers and practitioners, including organisers, participants and supporting specialists, still active after the event. It is recommended that such interdisciplinary workshops are used to build capacity to tackle complex challenges, such as risk communication, but require further testing. Insights into the design and implementation of such interdisciplinary workshops are given (e.g. team design, use of preparatory materials, and engagement of specialists and local stakeholders are presented), including critiques of challenges raised by the workshop participants. Guidance is provided to those interested in applying a Pressure Cooker approach and further adaptations of the approach are welcomed.

## 1. Introduction

Risk communication for Disaster Risk Reduction (DRR) is inherently interdisciplinary, requiring integration between social and environmental sciences, communication design, engineering and media, to name but a few. Not only does effective risk communication require interdisciplinary working, but it must also function across sectors, including government, industry, and academia, in addition to being based on a pragmatic understanding of end users. The challenges of interdisciplinary communication are especially pressing in the fields of risk and uncertainty (Klinke and Renn, 2012, 2014; Pappenberger et al., 2013), and have produced a wide, but fragmented body of literature on how communications are understood by the public





(Gigerenzer et al., 2005; Handmer and Proudley, 2007; Morss et al., 2008). More attention is needed to understand the communication of uncertainty among different professional groups (Faulkner et al., 2007). Despite a large amount of research interest, there remains relatively little consistency or formal agreement across fields as to how risk communication should be addressed (Demeritt et al. 2011; Ramos et al., 2010). While interdisciplinary literature reviews (Pappenberger et al., 2013, Carr et al., 2018) may address this problem to some extent at an academic level, direct collaboration allows for a more practical understanding of the working methods employed by experts from other disciplines (Bostrom, 2014; Drake et al., 2014; Fischer et al. 2011;). Interdisciplinary research comes with its challenges, but is seen as a necessity to tackle challenging societal problems in new interesting ways (Metzger and Zare 1999; Tobi and Kampen, 2018). Although there is a growing body of data and expertise in understanding, monitoring and predicting risk from environmental hazards, much of this data and expertise does not reach the individuals, communities and organisations who can use it to communicate and manage risks (Cook and de Lourdes Melo Zurita 2019). These investments in monitoring, research and capability can only achieve their full potential value if disaster risks are communicated effectively, empowering individuals and groups to pursue the mitigation, preparedness and response options that are best for them (Palenchar, 2008; Coombs, 2010; Fischhoff et al. 2011; Kasperson, 2014; Griffin et al., 2012; Miller et al., 2015). However, effectively communicating risk information to affected communities and local-level stakeholders is a significant challenge facing by researchers and practitioners, including those in Mexico – where the workshop and case studies in the paper were based.

The upcoming generation of young professionals and early career researchers have the potential to break down these barriers and work collaboratively from the outset on risk communication. Furthermore, this generation is experiencing the changing dynamics of communication technology with the potential engage with innovative solutions in risk communication (e.g. participatory risk mapping, Gaillard and Pangilinan 2010), or crowdsourcing flood data e.g. Le Coz et al. 2016. However, traditional academic education rarely provides opportunities to work across disciplines, to learn from peers from different educational, geographical, and professional backgrounds. In an earthquake risk communication workshop in Istanbul, it was seen that scientists from different disciplines, citizens, politicians, planners etc. had better exchanges for addressing risks collectively than they might have done individually (Ickert and Stewart, 2016). New training and capacity building is needed to develop applied tools and techniques that integrate knowledge and engage communities.

To address the above challenges, an interdisciplinary capacity building approach was developed by the Water Youth Network (WYN) and the Global Facility for Disaster Reduction and Recovery (GFDRR), with support from Natural Environmental Research Council (NERC), FM Global, and NASA focusing on risk communication following an intense 24 hour format ('Pressure Cooker'). The pressure cooker was implemented at the 2018 Understanding Risk Forum in Mexico City on May 14th and 15th, 2018. The event brought together 35 young professionals and researchers from different disciplinary backgrounds to innovate effective risk communication strategies based on real world case studies in Mexico. We define a Pressure Cooker event as a problem solving exercise where interdisciplinary teams need to devise a solution to real world



challenges within a 24-hour timeframe based on pre-prepared materials. This paper presents the Pressure Cooker approach developed to build interdisciplinary risk communication capacity in young researchers and professionals. It explores the impact of the workshop on the organisers and participants, and highlights the lessons learnt during its design and implementation, thus offering guidance to those interested in applying or adapting the approach.

**2. Interdisciplinary Problem Based Learning**

The Pressure Cooker approach is an example of problem based learning (PBL), with teams being given a challenge to solve in a fixed amount of time. PBL is used across a range of sectors, disciplines, and levels of education and training (De Graff and Kolmos, 2003; Harmer, 2014; Lehmann et al., 2008; Kolmos, 2009)) and originated in medical education (Barrows, Tamblyn 1980). PBL provides scope for collaboration across disciplines, within a team environment (De Graff and Kolmos, 2003), but

the implementation in the Pressure Cooker took place within a very concentrated time-frame. In this regard is shared features within another popular contemporary group teaching and team building technique – the hackathon (Briscoe, Mulligan, 2014). The social and team building opportunities afforded by a PBL/hackathon approach to building risk communication capacity are especially necessary given the interdisciplinary nature of the challenges facing risk communication in a geohazard context.

Workshops with a PBL and hackathon focus are not new to the area of geoscience and/or risk communication and the scale and focuses of past events have varied considerably. Recent events have included, e.g., data and technical focused hackathons at fixed locations, such as *Crisis Hack* and *Geo Hack Day* (Geovation, 2018, and GeoHackDays, 2015, respectively); and dispersed international events, with huge numbers of participants (>15,000) across the world engaging with the same event such as the *Space Apps Challenge* and *Shelter Urban Thinkers Challenge* (NASA, 2018, and Shelter 2015 respectively). As

will be seen, the Pressure Cooker aimed to combine the advantages of diverse international collaboration with the benefits of hackathon-style co-location.

A good problem is essential to good PBL (see Hallinger and Edwin, 2007, chapter 3 for considerations from the field of management). Given the context of the workshop, it was possible to involve a range of experts to generate detailed, real world

case studies (see section 3.2). It should be acknowledged that familiarity and direct experience can impact decision making, planning, and interpretation around risk and uncertainty (see, for example, St. John et al. 2000, Mulder et al. 2017 – but also counter-examples such as Nadav-Greenberg et al. 2008). As participants were drawn from across the world, and across disciplines it would have been extremely difficult to take this into account for scenario design if trying to 'even-out' any impacts. One option could have been the generation of fictional problems – potentially quite extreme ones. However, as the

US CDC's Zombie preparedness initiative demonstrated, novelty and high engagement with fictional scenarios may not translate into long term change (see CDC, 2017 for the original campaign, and Kruvand, Silver, 2013, Kruvand, Bryant 2015, Fraustino, Ma, 2015 for follow up research on its effectiveness). While the CDC campaign was aiming to affect change in end





users, not build practitioner capacity, it indicates that some caution is needed. With the workshop being held in Mexico city, local concerns were a natural choice and team make-up was finalised to attempt to evenly distribute expertise and familiarity with local contexts across groups.

### 3. Pressure Cooker Approach

5   This section outlines participant selection, case study briefing material, the pressure cooker workshop process, and the evaluation methodology.

### 3.1 Participant selection and teams

In line with the collaborative and interdisciplinary focus of the pressure cooker, participants were selected with a mix of skill sets and expertise, to foster peer-to-peer cross-disciplinary learning.

Upon receiving the applications (440 applicants from 74 countries), WYN reviewers scored applicants based on previous professional and academic experience and motivation. Remaining applicants were then assessed by the representatives of funding organisations (FM Global, NERC, NASA) and the WYN review team. All suitably scoring self-funded applicants were offered a place. The preliminary list of applicants was reviewed for regional, gender and disciplinary diversity.

15   Unfortunately, funding constraints (i.e. sponsor's focuses on specific geographical origins) presented an obstacle in enabling participation from African youth.

35 participants (14 self-funded) were able to join the pressure cooker, including three participating remotely via WhatsApp – 21 female and 14 male. The participants represented 13 different countries (Figure 1a). Further details of the participants

20   background can be found in Figure 1b.

Participants were allocated to one of five teams. In order to ensure disciplinary balance, each team comprised of at least one social scientist (including participants with community engagement skills), one environmental scientist, one modeller or engineer, one media and communication specialist and one design/ creative specialist. In addition, teams were composed to

25   maintain a mix of gender and regional backgrounds.

### 3.2 Case study briefing material

During the design of the Pressure Cooker, an early decision was made by organisers and the Steering Committee to focus on Mexican case studies. This was a pragmatic decision, taking into account that the event was taking place in Mexico, but also due to Mexico being a country exposed to multiple environmental hazards (e.g. floods, volcanoes, tropical storms and

30   hurricanes). Furthermore, organisers wanted to ensure that case studies were co-created with local stakeholders, in order to





present participants with real-life problems that require interdisciplinary solutions (see also the background of PBL in section 2.0).

Organisers reached out to a number of specialists (i.e. consultants and researchers) working on DRR issues in Mexico. An online session was organised in which the rationale of the project and a need for local case studies were presented; followed by a brainstorming session in which possible case studies were selected. Based on this, two detailed, real world case studies, relevant to local Mexican multi-hazard scenarios were selected. One was based on Iztapalapa, one of Mexico City's (CDMX) 16 municipalities, the other on Dzilam de Bravo, a coastal municipality in Yucatan state. A short description of the case studies is given in Table 1 and the full information can be accessed in the event's evaluation report (Water Youth Network, 2018).

A specialist was assigned to each case study, supporting the WYN team to: i) collate  background materials, ii) develop the case study documentation for participants, and iii) establish a working relationship with local DRR stakeholders: for Iztapalapa with representatives from the local civil protection agency, and in Dzilam De Bravo, with a researcher working in the region. To ensure each team was working on a different focus, five specific sub-challenges were developed with the case study

specialists. Within a specific case study, challenges were differentiated based on the hazards faced or a specific target group, but all needed to consider the multi-hazard context. Specific challenges were;

1. Iztapalapa – households at risk of flooding,
2. Iztapalapa – households at risk of building fracturing,
3. Iztapalapa – households facing resettlement,

4. Dzilam de Bravo – households dependent on fishing,
5. Dzilam de Bravo – communicating risk to schoolchildren.

### 3.3 Workshop process

Working in interdisciplinary teams, participants had 24 hours to develop a risk communication strategy in response to a set brief, then submitted for judging by an interdisciplinary panel of experts - as shown in Figure 2. They were asked to follow a three-step iterative process which formed three broad working sessions throughout the day: 1) understanding the risk context and audience at risk, 2) identifying the expected outcomes and impact of the proposed risk communication strategy, and 3) detailed development of a risk communication strategy. These steps were developed by the organisers and core mentors based

on their experiences of approaches to risk communication challenges. Three main sources of literature were used 1) guidance on audience analysis (JHU-CCP, 2016), 2) the UK Environment Agency guidance on understanding and communicating flood risk (Environment Agency, 2012), and guidance from BBC Media Action on selecting communication channels and using a theory of change approach (BBC Media Action, 2018). The importance of understanding the characteristics of the audience at risk and tailoring communication strategies to influence behaviour change was a central focus of the approach.





As participants were coming from a diversity of disciplinary and cultural backgrounds, a creative networking event was run the evening before the event. It allowed participants to socially interact and for organisers and supporters to observe the interaction between participants across disciplines. Participants were provided with a range of art and craft materials, asked to

make something that represented what risk communication meant to them, and presented their creations by discipline – examples are shown in Figure 3. While it is difficult to formally assess the outcomes of such an abstract task across disciplines, participants showed a relatively common understanding of risk communication by highlighting the importance of focusing on people: keeping the messages simple, clear, and easily understandable.

The Pressure Cooker workshop began at 9am, and ran continuously through to presentations at 9am the next morning (Figure 4). After dividing participants into respective teams, the workshop began with a general introduction on risk communication, moving on to expert presentations on the two case studies, before the case study and briefing material was handed out to the teams. Participants were not aware of the case studies or challenges prior to the event. The event was run in English primarily, with some translation from Spanish where relevant for guest local experts.

All participants were shown pre-recorded video interviews (with English subtitles) of members of the communities within the case studies, and transcripts of the interviews were provided (see example links in Water Youth Network, 2018, p.8). Each team was given digital and print copies of the detailed case study report (as detailed above), a shorter case study summary, copies of their specific brief, the key presentation slides, links to further data sources, and a notebook in which to record their

progress and decision making process. All of the briefs began with variations on the template phrase:

*Your team has been hired by local government office in [region] for a research consultancy to come up with a risk communication strategy including outputs targeted at [brief specific group/hazard] of [region]. As a part of your communication strategy you will need to develop an output tailored specifically for a selected vulnerable sub-target group.*

Each team was coached by an early career researcher to support them as required and stimulate discussion throughout the workshop and help them follow the three-step process. Teams received feedback from case study specialists, ensuring that the solutions developed were informed by the state-of-the-art science/practice and took into account local needs. At 4am all teams

had to submit a four-page overview document reporting on their project for submission to the judges. The documents had to cover the context, the communication strategy itself, and the intended outcome and impact of the proposed strategy. 24 hours after the start of the workshop, each group gave a 10 minute presentation to the judges, participants, and organisers, followed by questions. Participants were further required to submit notebooks showing the development of their project and key



decision-making points. Table 2 presents a brief overview of the communication strategies developed and the links to each team's submissions can be found in the event evaluation report (Water Youth Network, 2018, p.10).

The judging panel included; GFDRR, BBC Media Action, NERC, British Geological Survey, FM Global, National
Autonomous University of Mexico, and Iztapalapa Municipality Department of Civil Protection. Teams were judged on the following criteria: 1) decision-making process, 2) identification of expected outcomes and impact, 3) appropriateness of output for target audience(s) and aims (outcomes and impact), 4) originality, creativity and innovation in risk communication, 5) clarity of documentation and presentation, 6) applicability of the risk communication strategy.

The day after the judging, a field trip to the Iztapalapa Department of Civil Protection was organised, reinforcing the real life context of the challenges. While it could have been useful to include such a trip into the main Pressure Cooker, it would have given an advantage to groups working on the Iztapalapa case study and imposed further logistical limitations on an already busy event schedule.

**3.4 Evaluation methodology**

The event was monitored by two observers (members of the organising team), who conducted participant observation throughout the core 19 active hours of the Pressure Cooker. Observers took notes following a set of pre-described criteria: interdisciplinary working, strategic decision making, and engagement with feedback. Teams were asked to select a note taker to write down key decisions made, which was then fed back to the observers. The observers convened a feedback session with
all of the coaches after submission. The event judges evaluated the teams' outputs in a closed session, following input from observers and coaches. There was an open feedback session where all involved in the event came together to reflect and draw lessons for further events on the process and outputs. Post-event evaluation was carried out through two online surveys. The first, just after the event, covered: i) rating the overall experience, ii) learning about risk communication, iii) development interdisciplinary team skills, and iv) the likelihood to apply learning from the event (see evaluation and Figure 5, below). The
second survey, eight months after the event, collected examples of impact on risk communication knowledge, interdisciplinary working, and community building.

**4. Evaluation and (pathways to) impact**

The Pressure Cooker created new pathways to impact by building participants and organisers awareness and understanding of
risk communication and interdisciplinary working. The evidence collected from the feedback survey suggest that this is just the beginning of further impact, since the participants were so positive about the usefulness and applicability of what they



learned (see Figure 4). Additionally, organisers developed knowledge on how to design and implement interdisciplinary events and a new, interdisciplinary, network of young professionals and researchers has been generated aiming to achieve longer-term impact.

### 4.1 Participant feedback on risk communication/ interdisciplinary working

### 4.1.1 Participant reflection on risk communication knowledge generation

The post-event survey showed that participants reported gaining knowledge on their understanding of risk communication and how to approach it differently in their work or research. Many participants were not previously familiar with the different aspects that needed to be considered or the range of possible communication outputs. One social science participant indicated that he learnt about different ways of communicating risk from his team - "orienting and sensitising people about disaster
risks by leveraging cultural and traditional practices, this is something that I always had in mind but it is through this event that I learnt how to possibly do it." (social scientist). The participants valued learning about risk communication through real case studies, making it easier for them to think about how to apply the learning to their own context. Furthermore, they recognised the importance of connecting knowledge from the social and natural/engineering sciences.

One of the most apparent results from the feedback was participants' recognition of the importance of placing the target audience – people – at the centre of any risk communication (even though this was a feature that came out strongly in the ice breaker exercise). One environmental scientist participant highlighted how she is applying this to her own work – *"now from this experience, in my work we are already generating actions that fix our eyes on the most vulnerable groups and we are evaluating our current risk communication to locate areas of opportunity"* (environmental scientist).

Another powerful example of this is shown by a social science participant who has since had the opportunity to work on developing risk communication products for the National Meteorological Service of Argentina. He indicated how he has "been trying to cultivate a shared understanding of the problem" and bringing together an interdisciplinary team that includes "communicators and sociologists to focus on users' understanding of weather related information" which has resulted in
"forecasts and nowcasting products that are now involving users such as the National Civil Protection and the National Institute for Water." (social scientist).

### 4.1.2 Impact on participants for interdisciplinary working during and after the event

Participants learnt about the process of working with different disciplines, understanding their language, and taking into
account differing perspectives when making decisions as a team e.g. *"I would try to maximize the specific skills of every team member distributing more focused tasks"* (engineer). The pressure cooker had aimed to require a balance between individual



skill-sets. In the end, it was found that social sciences took the lead in comparison to natural sciences and design/communication. This was reflected in the learning reflections from the more technically oriented disciplines e.g. *"being a technocrat myself, I have started paying heed to the social science aspect of DRR, which perhaps started after the event"* (environmental scientist).

Participants also learned to be open to solutions outside of their discipline, as highlighted by one social scientist participant - *"I think that the best of all was to see how my colleagues thought about the different possible solutions. There were different approaches (technical, artistic, mass communication, etc.). I that sense I think that I learned some different ways to approach a problem.* " (Social scientist). However, the time pressure made it difficult for them to adequately consider/include the skills

of each discipline and when push came to shove - some ideas were dropped: "*Sometimes it was frustrating as the project ideas were not always relevant to my expertise but I guess that was the point of the exercise.*" (Environmental scientist).

## 4.2 Organisers experience

### 4.2.1 Organisers reflection on risk communication knowledge generation

Within the Pressure Cooker/PBL context, both the outputs of the workshop (Table 2) and the collaborative process which the organisers designed and participants were involved in, played a role in generating risk communication knowledge. From a capacity building perspective, the longer term applicability of skills and process learned are of greater significance. The organisers found that the three-step process (Figure 2) helped to structure the development of participants' ideas during the workshop, however, participants did not cover all three stages evenly. The judges pointed out a lack of detail shown in the risk

communication outputs, which was likely linked to observers' reporting a lack of time spent by teams on developing the communication strategy specifics (step 3 of the process), compared to understanding their audience at risk (part of step 1). While this is clearly not ideal, within the time constraints, the process, not the specific strategies, were likely more important for skills development. This was highlighted by participant feedback, who were challenged to think in detail about users' needs, which was a key learning experience. While participants were given access to data on the hazards in each area and had

at least one team member with technical skills, no team decided to focus on scientific or technological specifics when developing their solutions which some judges found disappointing - *"There was a strong focus on social science, we were expecting more hard science. It was refreshing to see the focus on people, but you can focus that with a few facts even if they are just basic science facts to build credibility."* *(Judge)*. This may have been due to the short time frame and data provided, and the end focus on the challenge on risk communication which was seen as more of a social science focused challenge. In

addition, more general participant feedback suggested that group dynamics and the Pressure Cooker environment may have played a role - *"It would have been helpful to have a reminder to keep thinking from the perspective of your own discipline.*" (Media and journalism specialist). This was an aspect that coaches could have done more to encourage within the teams.





It should be noted that while considerable resources were developed to give user context to the participants, due to practical constraints, none of the final outcomes were tested with end users. Nevertheless one of the case study specialists reflected on how the process of developing the challenge helped him and the local stakeholders "*delve further into the factors affecting*

*their successes and remaining challenges in risk communication ... it was that study itself and what the Pressure Cooker participants did with it in their limited time that gave me further insights into risk communication and its application*" (case study specialist Iztapalapa). It was also notable that all of the teams focused on women as the target group for their communication output and emphasised the role of community engagement.

**4.2.2 Organisers reflection on challenges when designing interdisciplinary events**

As mentioned, judges observed a relative lack of engagement with the physical science aspects of the case studies. One suggestion to enable more input from the natural/ physical sciences and its translation into the final solutions, was that teams could initially be discipline focused, then later split into interdisciplinary teams. While the structure of this specific event (with multiple challenges and limited time) would have made this difficult, it is worth further testing in longer events. In the design

of any event a balance needs to be struck between time and intensity to get the most out of individual skills/disciplines. While step 2 of the process encouraged teams to think of a range of ideas, more emphasis on broad exploration may have resulted in other approaches with more physical science engagement.

When assigning a challenge brief, the organisers learnt to manage expectations and clearly indicate what is expected as a final

outcome. There is always going to be a tradeoff between enforcing a template and providing flexibility to enable more openness and creativity. Another challenge is deciding the level of detail to provide on the case study, which practically depends on the ability of the team to have access to local contacts and information. Feedback from case study specialists and topic specific specialists proved useful for participants and was feasible because of the UR2018 conference context. Although a field trip was not possible during the event, the video interviews were deemed useful. A limited number of interviews cannot encompass

all viewpoints, however – where possible a field visit or other, more direct, engagement is recommended.

While the pressure cooker emphasised interdisciplinarity, it is important to ensure individual expertise and specialisations are utilised. Interdisciplinarity should aim to enhance the applicability of specialisations, not dilute them to generalisations. Despite the success of the ice breaker before the Pressure Cooker to help participants get to know one another and their background in

a creative way, one participant reflected afterwards: *"If we had more time to get to know each other we might have challenged each other more"* (Social scientist). As it was, the focus was more on encouraging participants to challenge the case study problems (a useful brainstorming exercise), not each other. Having the confidence to appropriately challenge professionals from other disciplines is an interdisciplinary skill in its own right – arguably one that does not develop in just 24 hours. Thus,





such a networking event between organisers, participants and supporters is strongly recommended as an integral part of interdisciplinary events.

All organisers' time in preparation and execution of the event was in-kind/ voluntary. In some cases, this may not be feasible and the capability of WYN volunteers was central in facilitating this event. Although the WYN volunteers put a lot of time into the event, it was rewarding for them to gain experience in organisation but also knowledge on interdisciplinary working and risk communication, as indicated by one organiser "*It helped me to gain experience in coordinating, mentoring and communicating with people from different backgrounds which are skills that I can now put in practice for other events. I think I'm more aware of the importance of interdisciplinary working for my future career & how it's going to become a really key & unavoidable aspect of working in academia in the near future*" (WYN organiser).

### 4.3 Extending the pathways and impact after the event

The event built a new network of researchers and practitioners interested in furthering risk communication knowledge. This has, and is expected to continue having, an impact on participants' ability to exchange knowledge and collaborate – hopefully with a longer term rippling effect on relevant wider research and practitioner communities

### 4.3.1 Supporting community building before and during the event

The steering committee brought together young professionals and experienced professionals from research and practice. The conference context ensured an informal atmosphere throughout the event, and the creative networking activity helped to set the scene for getting to know each other. Having a youth-led organising team meant that the participants felt at a similar level to them and could interact easily. However, the experienced professionals/mentors/funders were approachable and non-hierarchical, creating an interactive and supportive environment.

While the 24-hour event forced intense collaborative working, the wider conference context allowed networking to continue during the main event and evenings. A WhatsApp group was created before the event which was used to facilitate interaction before and during the event. The powerful impact of the new network generated was very apparent in the feedback survey, a reflected an engineer participant - "*meeting like-minded young professionals who care about reducing hazard risk and who are also hard-working, really intelligent, and super fun was so refreshing and encouraging.* " (Engineer).

The value of generating such a network was recognised by the organisers as the key output of the event, (albeit unintentional!): '*I have been utterly astounded by the level of enthusiasm and connectivity of the community that participated in the event. Although not one of [sponsor name contained for anonymity] aims when we agreed to fund this event, I believe it's biggest*



*success has been in building a community of young researchers who are aware of the demand for their knowledge and skills in the disaster risk reduction arena - and that should they continue to a career in academia, that they continue to bridge worlds between universities and practitioners.*' (sponsor representative). The participants reflected the importance of generating this network - *'The network of other outstanding young professionals that I met during the Pressure Cooker event has been amazing*

*- I think probably the best take away beyond any knowledge or single lessons learned.'* *(social scientist).*

### 4.3.2 Maintaining and widening the network

Strengthening this intergenerational network and the peer-to-peer network is very important to ensure the continuous impact of the event. To maintain sustainable engagement of the participants, the WYN offered all participants to join the WYN DRR

team; to date, six participants have become active members and are actively involved in follow-up projects. All participants are now members of the UR community. However, on a more informal level, the WhatsApp group remains very active for sharing opportunities to further engage in person (e.g. additional conferences), knowledge (e.g. articles, training material) and career opportunities.

The lessons learned on interdisciplinary working are expected to be applied to future events and projects, which will influence how broader disciplines are engaged in the topic of risk communication. For example, to promote engagement of the design community, which was found to be challenging (only 9% of applications received were from those with a design background), an article was prepared about the event for the European Academy of Design (2019) conference (see Lickiss, Cumiskey, 2019 forthcoming), in combination with new teaching collaborations outlined below. Furthermore, ten blog posts were published

by a combination of organisers, supporters and participants for a mix of academic, practitioner and policy-based audiences. For example, an organiser wrote in the BBC Media Action Insight blog (BBC Media Action, 2018), jointly a participant and mentor wrote in the British Geological Survey Geoblogy blog (BGS, 2018) and a participant wrote in their own NGOs blog (NexoDRR) explaining their solution (Nexo, 2018).

A limited amount of funding was available to participants wishing to develop follow up events or outreach activities on risk communication in their respective localities in collaboration with the WYN. One outcome of this is the upcoming WYN Hack-the-risk event co-organised with the Columbian participant and similarly supported by FM Global (Hack the Risk, 2019). The WYN will take forward lessons on interdisciplinary working. Furthermore, many of the participants have since reunited a conferences e.g. Annual Geoscience Union, European Geosciences Union and are further organising interdisciplinary sessions/

events on risk communication e.g. Royal Geographical Society Post Graduate Forum (August 2019) session, UR Field Lab risk communication track in Chiang Mai (June, 2019), American Geophysical Union (AGU) workshop on understanding your audience (December, 2018). Furthermore, the organisers/supporters have collaborated across different academic institutions





e.g. teaching collaborations between the University of Reading with the Environment Agency and Met Office, and between GFDRR/ Understanding Risk and the University of Singapore.

## 5. Way forward - lessons and guidance for future applications

The application of the Pressure Cooker approach described in this paper presented a novel method for advancing the environmental risk communication agenda, an increasingly important field of study and practice. Our approach aimed to build a case for the importance of interdisciplinary, intergenerational, and user-centric approach to risk communication. The need for such an approach emerged through our understanding of a deficit of current risk communication approaches, that are often one-way communication and based on a deficit-model (Cook and de Lourdes Melo Zurita 2019).

The experiences described show that Pressure Cooker approach has potential as an interdisciplinary capacity building tool for risk communication. Here we take stock of the main lessons to serve as a pointers for future Pressure Cooker inspired workshops.

- *Process vs. output:* One needs to be realistic about the technical sophistication of the outputs given a short, 24-hour time frame. Rather than expecting finalised products, the process should be seen as the key opportunity to learn about risk communication and interdisciplinary working. However, if more detailed and technically-sound solutions are expected then a longer time-frame should be ensured.
- *Utilise all disciplines*: In an interdisciplinary setting, there is a danger of not sufficiently integrating insights from different disciplines, thus different formats should be explored. For example, grouping participants by disciplines to develop detailed solutions once a broad approach has been reached. Interactive networking between participants should be encouraged.
- *Real-life case studies:* Employing real-life case studies was crucial to the process, since it allowed participants to understand the context and develop solutions based on local needs. However, where possible a case-study visit prior to the event is recommended.
- *Diversify organising/supporting team:* The involvement of senior mentors, case-study specialists, topic-specific specialists, peer-peer young professional mentoring/organisation was widely valued. This enabled real-time feedback in a non-hierarchical environment embracing intergenerational knowledge exchange and networking. Variants of the interaction frequency and duration between participants and supporters can be further tested. Enabling end users to be directly included in the pressure cooker should be encouraged where logistics permit.

Next, some practical questions are presented as guidance for those interested in organising a similar workshop.

- *Is the event stand-alone or it will it be a part of an already ongoing workshop or conference?* This could determine the availability of senior mentors as well as logistical elements (e.g. availability of a venue).





- *What is the feasible/ desired length of the event?* As shown, there is a trade-off between the event length and quality of outputs which must be considered early in the event design.
- *Does your team have enough human resources and time to organise the event?* The event proved to be time-intensive for organisers, but maximised voluntary contributions.
5 - *Do you have an outreach strategy in place to target high-quality participants from different disciplines?* Utilising an existing network proved useful to generate a large number of high-quality applications, nevertheless gaps existed in reaching some disciplines.
- *Do you have access to funders for participants?* The event targeted, with some limitations, geographically diverse participants, thus funding availability was crucial. Without funding resources there is a danger that those from less 10 developed countries cannot attend. The incentives for possible sponsors should be identified.
- *Who can be mobilised within your network as mentors, local stakeholders and topic-specific specialists?* The experiences showed that these are of crucial importance in the process and should be mobilized as early as possible to co-design the event, especially case studies.

## 6. Conclusions

Environmental risk communication is a complex challenge, requiring interdisciplinary approaches. Here we developed a Pressure Cooker event as a problem solving exercise where interdisciplinary teams devised solutions to real world challenges in 24-hours. A three step process helped to break down the complexity and guide participants. Interdisciplinary capacity building approaches are necessary for the next generation to build their skills, knowledge and capabilities. Despite some 20 challenges, overall the Pressure Cooker approach has been shown to be as a useful way to achieve this. Participants increased their risk communication knowledge and interdisciplinary teamwork skills, organisers learnt lessons on designing such an approach, and an interdisciplinary community of peers and seniors was generated. The long term impact has yet to be seen, although there are some early indications that it will be realised, but is expected to have a rippling effect throughout the coming years. Putting together an interdisciplinary event is not easy, but offers a range of benefits once realised. We strongly encourage 25 more applications and adaptations of the Pressure Cooker approach for risk communication and it is likely to be applicable to other areas. Further applications will help to develop guidance on which approaches, or combinations thereof, to use for different contexts, target audiences, or problems.

## 7. Acknowledgements

30 The authors would like to acknowledge the additional Water Youth Network organisers in particular, Gabriela Nobre, Miguel Trejo, and Nhilce Esquivel, along with the Understanding Risk GFDRR organising team in particular Simone Balog-Way, the sponsors for the event NERC, FM Global and NASA, and all the supporters of the event, as well as the participants themselves. The event was only made possible with their commitment before, during and after the event.





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





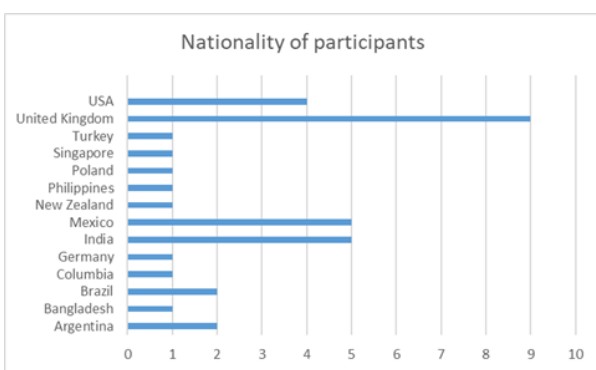

**Figure 1a: Nationalities of the participants**

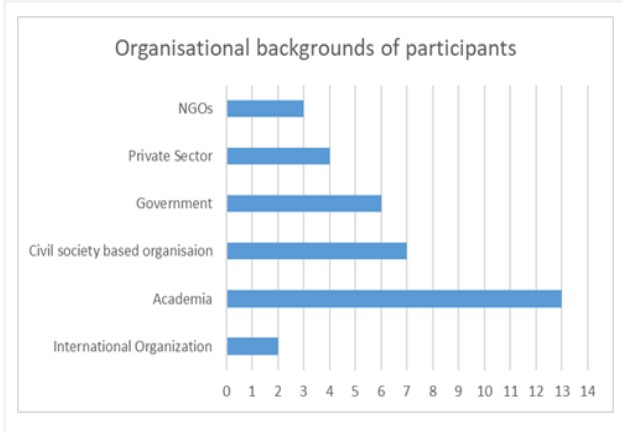 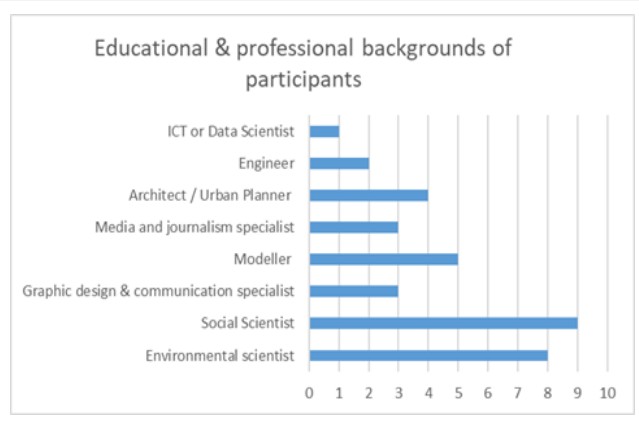

**Figure 1b: Organizational, and Educational and professional backgrounds of participants**





**Figure 2 – 24-hour Pressure Cooker event process**




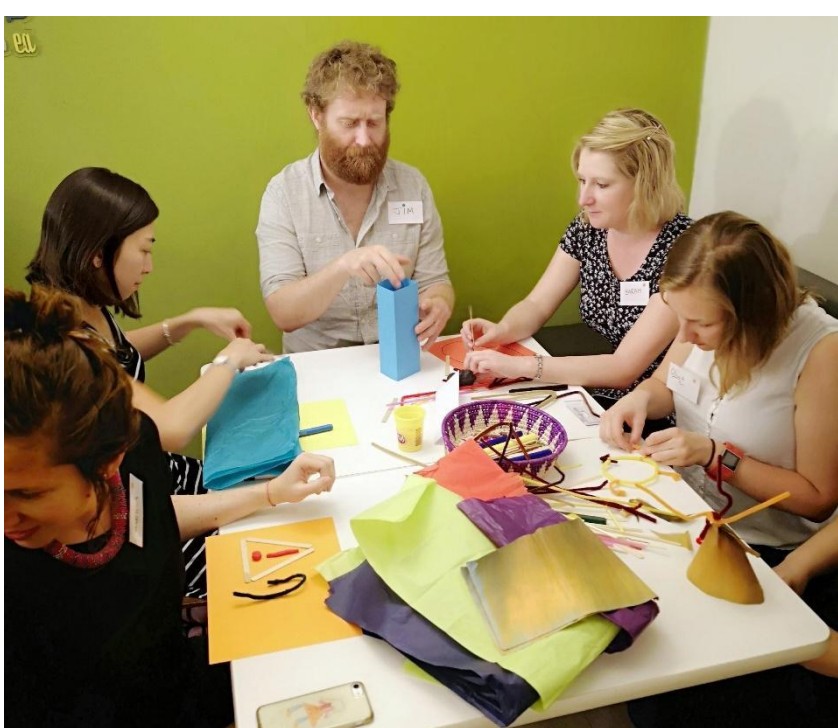

**Figure 3 – Creative networking in action with participants from mixed disciplinary backgrounds exploring what risk communication meant to them.**

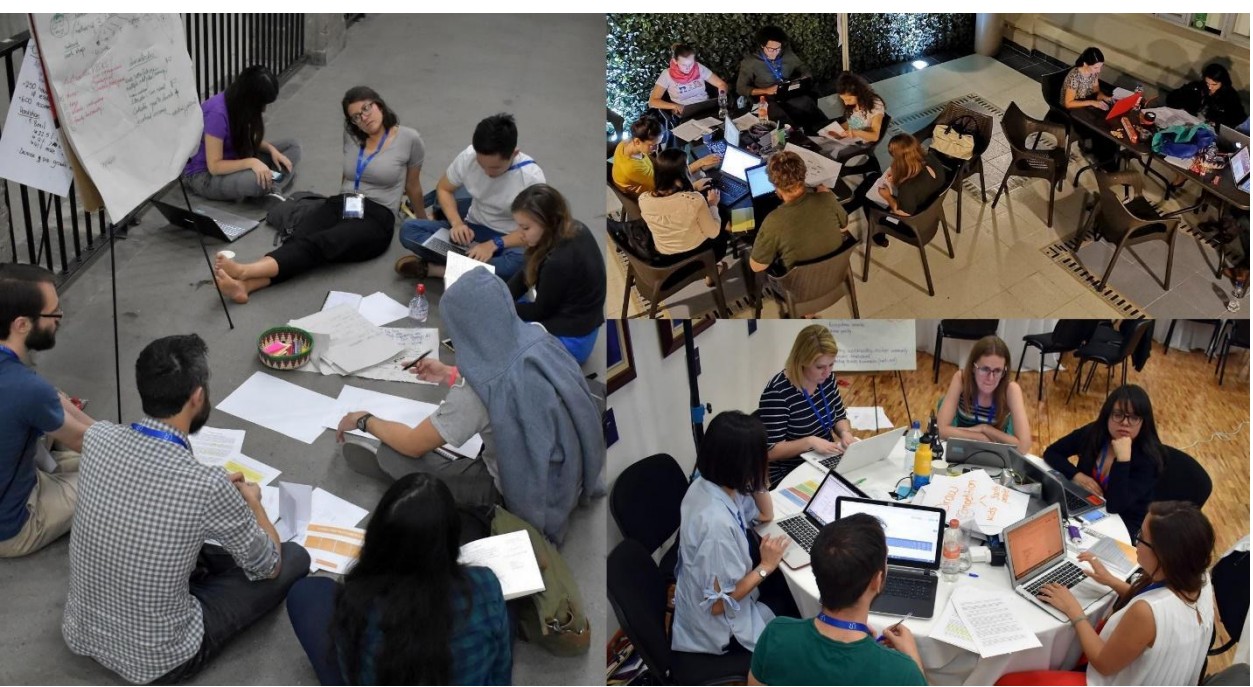

5 **Figure 4 – Teams in action throughout the day and night in a range of working environments.**





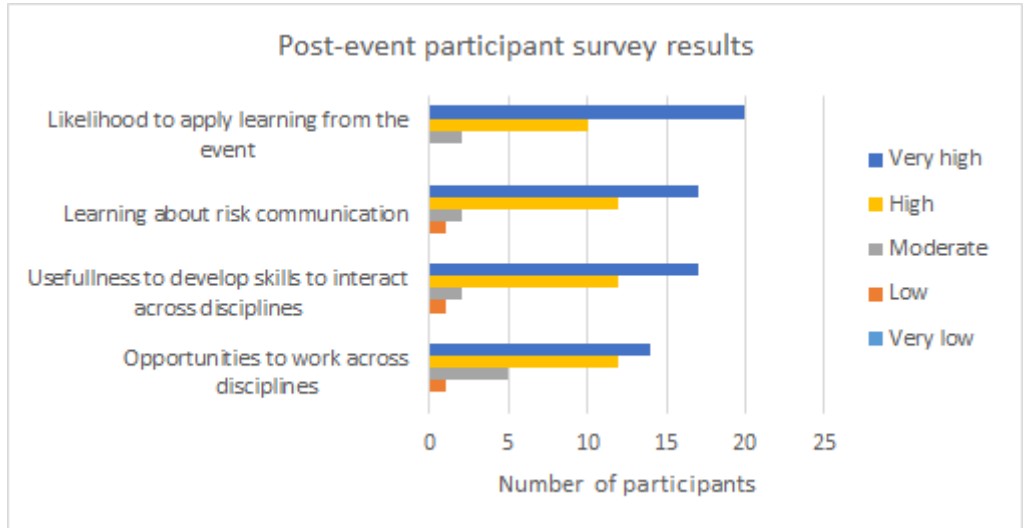

**Figure 5: Post-event survey results, asking participants to self-rank on working across disciplines and risk communication learning.**





**Table 1 Case study descriptions**

| **Case study 1: Iztapalapa, Mexico City** | **Case study 2: Dzilam de Bravo** |
|---|---|
| Located in the western part of Mexico City, with a population of nearly 1.9 million, Iztapalapa is one of the most deprived areas, experiencing a range of social (theft, gangs, high level of marginalisation, low access to health care, poor infrastructure) and environmental (deforestation, land use change, air pollution, excessive water extraction) problems.<br><br>The most common natural hazards are geological (high danger of faults, fractures and subsidence; medium danger of earthquakes, landslides and water erosion; low danger of volcanoes, mudflows and wind erosion) and hydrometeorological (high danger of floods; medium danger of strong winds, frost and hail; low danger of cyclones). The most affected groups tend to be those of socio-economic marginalised people living in the informal lands and land reclaimed from the lake. | Dzilam de Bravo is one of the coastal municipalities of Yucatan State, with a population of approximately 3000 people, out of which nearly 50% is in poverty. Dzilam de Bravo is located in a region highly impacted by hurricanes; in addition to cold fronts, tropical storms, droughts, forest fires and shoreline erosion. During hurricanes the area gets flooded causing economic and infrastructural losses.<br><br>The social vulnerability in Dzilam de Bravo is exacerbated by the high percentage of low-income population, thus families are living in houses with high levels of overcrowding and have few opportunities for access to health services. In addition, local disaster management plans and training programmes are non-existent. |





**Table 2: Summary of teams' results (edited tabulation of summary in Lickiss, Cumiskey, 2018)**

| Team | Target group | Risk communication output |
|---|---|---|
| Iztapalapa – households at risk of flooding | Primary school children | 'Water Ambassador Programme' at primary schools using existing community outreach methods, alongside a municipal programme to install water catchment tanks on houses. Ambassadors could become 'Guardians of the Drains' to help ensure that local drainage systems were not obstructed. |
| Iztapalapa – households at risk of building fracturing | Young mothers and ni-nis (youth not in education or work), children and elderly | Training programme for women to act as community ambassadors to co-design further community engagement activities. Initial activities would include training on repairing and preventing cracking, community mapping, increasing community awareness of the contribution of street garbage to flooding (which worsens fractures and subsidence). |
| Iztapalapa – households facing resettlement | Women (mostly informally employed) | Opening up iterative dialogues around the resettlement process, giving a sense of informed ownership in the decision making process. Activities include; town hall meetings, trips to proposed resettlement sites and analysis of potential risks at new sites. |
| Dzilam de Bravo – households dependent on fishing | Women (from fishing households) | Female community champions would engage with community, church, and sports groups to extend awareness of risks, facilitate dialogue and empower communities (including social media, children's activities, and community mapping visualisation). |
| Dzilam de Bravo – communicating risk to schoolchildren | Teachers and children (9-12 years) and indirectly to families | Overall winners – Teacher's guide to risk communication tailored for children, integrating risk communication into lessons across a range subjects, e.g. making a neighbourhood flood maps as a geography lesson. |

