# Peer review of "Interdisciplinary Pressure Cooker: environmental risk communication skills for the next generation"

_Geoscience Communication, 2019_

## Referee Comment (RC1) · Wendy Bohon (Referee) · 23 Jul 2019

The pressure cooker approach described in the manuscript is an interesting way to tackle a complex and persistent problem in risk communication. The authors have done a thorough job in outlining the planning, process and implementation of the workshop, and they have adequate evaluation results to prove its effectiveness. They also provide thoughtful and substantive lessons learned and guidance for those who want to implement this approach.

While I have some questions for the authors that stem from my curiosity (for instance, do they think it would be useful to allot an entire day before the pressure cooker to

allow the participants to get to know one another?) I have no significant comments or revisions to suggest. The only thing I would like to see explained in more detail is the time commitment needed from mentors and volunteers; this would be useful planning information for those looking to replicate this exercise.

In conclusion, this is a well constructed and well written paper outlining a useful skill-building and cohort creating process and I recommend that it be published.

---

## Referee Comment (RC2) · Anonymous Referee #2 · 30 Jul 2019

This paper tries to capture observations and lessons from a complex approach to enabling cross-disciplinary working. Although the premise of the paper is good the methods are a little vague and the analysis and results presented are not as robust as I would have expected. Overall, I believe that with some rewriting this could be a valid contribution to the journal however there are some issues that need to be addressed first:

Ethics: Despite this being research that involves human participants, and puts them under stress during a 19hr workshop, there is no mention of ethical clearance – should be in the acknowledgements at the very least. In addition there are moments in the

paper where it would be very easy to work out who the participants are e.g. page 8 line 21 and are the photos relevant or do they add value to the paper?

Consistency: There is a lack of consistency throughout the paper in regards to some key terms. Given that language is one of the critical barriers for interdisciplinary multi-actor working this must be addressed. Capitals are used then not used e.g. the term pressure cooker is presented in various forms, with capitals, without, with quotation marks and without (see page 2 line 31).

Young professionals: The paper explains that an aim is to bring young professionals together, but there is no definition of what this means. This is particularly problematic on page 4 lines 10-16 where the criteria for participant selection has no relevance to being a 'young professional'. It is unclear why this selection criteria has been identified e.g. page 4 line 11 – why motivation? Why a representation of funders? How are they scored? Why are they scored? Why now is there a geographical element? What has any of this got to do with young professionals?

You use the term applicants? But this is also not explained, why are they applicants? Did you have funding available to them to attend? Why not keep consistency are call them participants?

WhatsApp? The remote participants were mentioned in section 3 but then this was not reflected on in the analysis. Did the remote participation have no impact on how the workshop ran? How did these participants perceive their involvement? Can you analysis the results to demonstrate the difference between these remote participants and in-person participants? Multiple terms for different or the same actors – it is not clear

Within the paper the authors discuss the following actors but without definitions or clarity on their roles, I believe sometimes these terms are used interchangeably:

- Applicants, participants, young professionals, researchers, mentors, specialists, experts, steering committee, case study specialists, experienced professional, early career researchers….etc. - Who and why is someone a specialists or an expert and someone not?

'Real world' – please define what you mean by this?

Resources provided to the participants appear very UK focused was there a reason for this? Did it have an influence in the results? English language is mentioned but not in the analysis or discussion.

Ice breaker event review: What was the purpose of observing the participants during the ice breaker session? It is not clear? Did the participants develop or just demonstrate a shared understanding of risk communication – this brief analysis and conclusion of the ice breaker event seems flippant and adds little value to the rest of the paper. Influence of the intense environment

There is no mention of the 19hrs of intensive work on the results of the workshop. I would have thought that this would have been discussed in the results analysis. After all this is an analysis of the pressure cooker method – yet no mention of whether this was ethical, exhausting for participants, that their judgement was impaired….any negative side effects at all!??? E.g. page 9 lines 19 and 20….a lack of outputs not only a time issue but perhaps a tiredness issue? Lack of motivation?

Analysis I have already mentioned some areas of improvement for the analysis sections. However overall I don't believe the method of analysis is described in enough detail. How were the observations recorded? What was the data? How was the data analysed? Who did that work? What was their positionality?

In addition there is no mention of the influence of the context of the UNISDR Global Platform on participants and how they engaged? This should be included – was it mid platform, before or after this would have an influence on participant's motivations and knowledge.

Clumsy or wordy sentences which were difficult to understand: - Page 2 line 10. - Page 4 line 28 Confusing terms: - Page 3 line 22 A good problem. . . .what is a good problem?? - Page 8 Line 16 placing the target audience – people – at the centre. . ..No need to highlight the audience as 'people', unless there was a specific and relevant group of people under focus? - Page 8 line 4 risk communication/ interdisciplinary work. . .not / but 'and' - Page 8 natural/engineering. . ..as above, use 'and' Typos/spellings and mistakes: - Citations do not require detailed explanations within the brackets, remove these e.g.page 3 line 22 & page 3 line 31 and 32. Page 2: - Line 15 – faced not facing - Line 20 missing 'to' between potential and engage - Line 21 missing brackets after - flood data - Line 29 NERC should have UK before full name, so that international readers understand it is the UK research council - Line 30 NASA expand for constancy with other acronyms Page 3: - Line 10 is shared features?? - Line 30 US CDC? Page 4: - Line 4 include the word 'The' in subtitle and keep subtitles consistent in terms of using or not using capitals for each word. - Line 17 do not start a sentence with a number Page 5: - Line 7 CDMX?

―――――――――――――――――――――

---

## Author Comment (AC1) · 30 Aug 2019

Dear reviewers,

Thank you very much for your helpful comments that will allow us to improve the paper. Please find below an overview of how these comments would be taken into account in a revised version of the paper. The reviewer comments are numbered and the responses are written in italics below.

Yours sincerely, Lydia Cumiskey and the author team.

Reviewer 1 Comments: 1. The pressure cooker approach described in the manuscript

is an interesting way to tackle a complex and persistent problem in risk communication. The authors have done a thorough job in outlining the planning, process and implementation of the workshop, and they have adequate evaluation results to prove its effectiveness. They also provide thoughtful and substantive lessons learned and guidance for those who want to implement this approach. While I have some questions for the authors that stem from my curiosity (for instance, do they think it would be useful to allot an entire day before the pressure cooker to allow the participants to get to know one another?)

Authors' response: Thank you very much for this positive reaction to the article. It is definitely useful to allow participants more time to get to know each other, however, a full day may be too much unless it is part of the challenge in some way - for example - having the field visit, or doing preparatory group work to familiarise themselves with the topic or case study.

2. I have no significant comments or revisions to suggest. The only thing I would like to see explained in more detail is the time commitment needed from mentors and volunteers; this would be useful planning information for those looking to replicate this exercise. In conclusion, this is a well-constructed and well written paper outlining a useful skill building and cohort creating process and I recommend that it be published. Authors' response: We will add the following sentence to section 4.2.2 to address the time commitments: 'The up-front time commitment to organise the event from the core WYN volunteer team was around two months, with an additional week of work from the mentors and steering group members.'

Reviewer 2 Comments:

This paper tries to capture observations and lessons from a complex approach to enabling cross-disciplinary working. Although the premise of the paper is good the methods are a little vague and the analysis and results presented are not as robust as I would have expected. Overall, I believe that with some rewriting this could be a valid

Interactive
comment

contribution to the journal however there are some issues that need to be addressed first:

1. Ethics: Despite this being research that involves human participants, and puts them under stress during a 19hr workshop, there is no mention of ethical clearance – should be in the acknowledgements at the very least. In addition there are moments in the paper where it would be very easy to work out who the participants are e.g. page 8 line 21 and are the photos relevant or do they add value to the paper?

Authors' response: Thank you for the detailed feedback on our paper, we aim to address each of the points raised as follows: The event was a skills and capacity building event, accepted by the UR2018 conference, it was not run a research study (so the workshop itself did not go to a research ethics committee). This paper reports on that event, its approach, methodology and limitations. The application form for the event specified that it would be a 24hr event and participants were free to withdraw if they felt uncomfortable with this idea, during both the application procedure and the event itself. The event had a very high staff to participant ratio, to monitor the safety of participants throughout, including a mentor monitoring every team directly. In the post event surveys, participants were free to remain anonymous, to withdraw from the survey at any time, and were informed that their responses may be used to improve future workshops and/or reports and publications. Regarding the photos, it was clearly announced during the event that photographs were being taken and many participants shared their own photos to the event pool. Specific approval was sought/obtained from each of the individuals identifiable in the photos used in this paper. All those approached approved the use of the photos in this paper. While not all respondents to the post-event surveys chose to remain anonymous, we have, for consistency, anonymised all responses used directly in this paper. In the specific example you mention, where it would be possible to work out the identity of a participant, we have already checked with that participant directly with the response: "I don't mind, you can use my name or my answers as you need." (in addition to photo permissions). We would be happy to add a sentence or two

to the acknowledgements stating we have these permissions if the editors recommend that for this journal publication.

2. Consistency: There is a lack of consistency throughout the paper in regards to some key terms. Given that language is one of the critical barriers for interdisciplinary multi-actor working this must be addressed. Capitals are used then not used e.g. the term pressure cooker is presented in various forms, with capitals, without, with quotation marks and without (see page 2 line 31). Authors' response: Thank you for spotting this inconsistency. We have updated these points across the document (to 'Pressure Cooker').

3. Young professionals: The paper explains that an aim is to bring young professionals together, but there is no definition of what this means. This is particularly problematic on page 4 lines 10-16 where the criteria for participant selection has no relevance to being a 'young professional'.

Authors' response: All participants were 35 years of age or under. This was a specific criteria in the application - only those that were under 35 were then passed to the next round to be assessed based on the criteria mentioned. A sentence will be added to section 3.1 clarifying our use of the term 'young professional' in the initial screening process: 'Applications were open to young professionals, a term used here to refer to applicants under 35, working, researching, or studying in relevant fields (e.g., risk modelling, civil engineering, etc.).'

4. It is unclear why this selection criteria has been identified e.g. page 4 line 11 – why motivation? Why a representation of funders? How are they scored? Why are they scored? Why now is there a geographical element? What has any of this got to do with young professionals? You use the term applicants? But this is also not explained, why are they applicants? Did you have funding available to them to attend? Why not keep consistency are call them participants?

Authors' response: Given the years of experience the Water Youth Network have with

reviewing applications for capacity building events, we selected this 'motivation' criteria because it gives an indication of the extent to which the applicant may learn from the experience gained and use it in their work or study.

Yes the majority of participants were funded - as indicated on page 4 line 18. The funders had specific geographical preferences for the participants that they could fund. For example, NERC could only fund UK based researchers, and FM global could only fund participants in countries where they have active offices. This was outside the control of the organisers.

We use the term applicants and participants because they were not participants until they were selected (see also response to point 6). For example, Line 11 page 4 is referring to all applicants not just the 35 participants.

5. WhatsApp? The remote participants were mentioned in section 3 but then this was not reflected on in the analysis. Did the remote participation have no impact on how the workshop ran? How did these participants perceive their involvement? Can you analysis the results to demonstrate the difference between these remote participants and in-person participants?

Authors' response: The decision to involve the participants via WhatsApp was a last minute work-around to unexpected travel difficulties relating to visa and flight issues. If this had been a planned element of the workshop more emphasis could have gone into evaluating it. Therefore, only the participants physically attending filled out the survey. A sentence will be added to section 3.1 to clarify this: 'Due to last minute visa and flight complications, three participants engaged remotely via WhatsApp. This remote engagement was not a planned part of the event and was not captured by the pre-planned evaluation methods described in section 3.4 and 4.'

6. Multiple terms for different or the same actors – it is not clear within the paper the authors discuss the following actors but without definitions or clarity on their roles, I believe sometimes these terms are used interchangeably: - Applicants, participants,

young professionals, researchers, mentors, specialists, experts, steering committee, case study specialists, experienced professional, early career researchers. . ..etc. - Who and why is someone a specialists or an expert and someone not? 'Real world' – please define what you mean by this?

Authors' response: Applicants was used in section 3.1 to refer to all those who applied to take part in the pressure cooker before their attendance was confirmed (i.e., during the application processes). After that, they are referred to as participants and young professionals. The use of 'young researcher' in the abstract will be removed, with the remaining use of 'young researcher' being in a quoted feedback response.

Experienced professionals were used in the article to describe the group of more experienced individuals that supported the event in comparison to the young professionals. The distinctions between mentors, case study specialists was necessary as some of the experienced professionals supported on different things.

Thank you for spotting the use of expert. This has now been corrected to case study specialist. We made a conscious effort not to use the term expert.

Real world in this case means that a hypothetical challenge/case study was not used and that it represented an actual challenge facing society.

7. Resources provided to the participants appear very UK focused was there a reason for this? Did it have an influence in the results? English language is mentioned but not in the analysis or discussion.

Authors' response: By resources, if you refer to funding, then 7 of the participants were funded very generously by NERC given their interest in supporting this workshop.

The survey results from one participant mentioned their challenges to engage because of their limited English - however this is difficult to distinguish from the results. This was a criteria in the screening phase to ensure all participants were proficient in English.

8. Ice breaker event review: What was the purpose of observing the participants during

the ice breaker session? It is not clear? Did the participants develop or just demonstrate a shared understanding of risk communication – this brief analysis and conclusion of the ice breaker event seems flippant and adds little value to the rest of the paper.

Authors' response: As mentioned the participants were split into groups and had to make something that demonstrated what risk communication meant to them (i.e., directly relevant to this workshop) and then share/discuss it with the group. From this discussion, it was easy to see that all participants had a similar view on the importance of risk communication. While no formal analysis was conducted, this social event was used to help participants warm up and get to know one another before engaging in a strenuous team exercise requiring mutual understanding.

9. Influence of the intense environment: There is no mention of the 19hrs of intensive work on the results of the workshop. I would have thought that this would have been discussed in the results analysis. After all this is an analysis of the pressure cooker method – yet no mention of whether this was ethical, exhausting for participants, that their judgement was impaired. . ..any negative side effects at all!??? E.g. page 9 lines 19 and 20. . .a lack of outputs not only a time issue but perhaps a tiredness issue? Lack of motivation? Analysis I have already mentioned some areas of improvement for the analysis sections.

Authors' response: A few of the responses to the survey indicated that they struggled with the lack of sleep, however, this was outweighed by their positive response/ experience of the event. This was also why we recommend having a longer event if possible. However, further details on this aspect can be added to the paper.

10. However overall I don't believe the method of analysis is described in enough detail. How were the observations recorded? What was the data? How was the data analysed? Who did that work? What was their positionality? Authors' response: The observations (notes written by observers) are not formally analysed in this paper,

but input from the observers was used by the judges to decide the winning team (as mentioned in 3.4). Detailed analysis of the observations, along with working materials generated by the teams, may form part of a later fine scale multimodal motion analysis, but this is outside of the scope of this paper (and possibly this journal), which focuses on the event organisation, implementation, and feedback.

11. In addition there is no mention of the influence of the context of the UNISDR Global Platform on participants and how they engaged? This should be included – was it mid platform, before or after this would have an influence on participant's motivations and knowledge.

Authors' response: As indicated on page 2 line 31, the event was completed within the Understanding Risk event in Mexico not the Global Platform on DRR. This was completed during the first two days of the event while the side-events were taking place. This allowed participants time to attend the main conference event the following three days.

This can be clarified in the text.

12. Clumsy or wordy sentences which were difficult to understand: - Page 2 line 10. - Page 4 line 28 Confusing terms: - Page 3 line 22 A good problem. . ..what is a good problem?? - Page 8 Line 16 placing the target audience – people – at the centre. . ..No need to highlight the audience as 'people', unless there was a specific and relevant group of people under focus? - Page 8 line 4 risk communication/ interdisciplinary work. . .not / but 'and' - Page 8 natural/engineering. . ..as above, use 'and' Typos/spellings and mistakes: - Citations do not require detailed explanations within the brackets, remove these e.g. page 3 line 22 & page 3 line 31 and 32. Page 2: - Line 15 – faced not facing - Line 20 missing 'to' between potential and engage - Line 21 missing brackets after - flood data - Line 29 NERC should have UK before full name, so that international readers understand it is the UK research council - Line 30 NASA expand for constancy with other acronyms Page 3: - Line 10 is shared features?? -

Line 30 US CDC? Page 4: - Line 4 include the word 'The' in subtitle and keep subtitles consistent in terms of using or not using capitals for each word. - Line 17 do not start a sentence with a number Page 5: - Line 7 CDMX? Authors' response: Thank you for spotting these issues. These can be easily adjusted.